# Improving 10-deacetylbaccatin III-10-β-O-acetyltransferase catalytic fitness for Taxol production

Bing-Juan Li[1,2], Hao Wang[1,2], Ting Gong[1,2], Jing-Jing Chen[1,2], Tian-Jiao Chen[1,2], Jin-Ling Yang[1,2] & Ping Zhu[1,2]

The natural concentration of the anticancer drug Taxol is about 0.02% in yew trees, whereas that of its analogue 7-β-xylosyl-10-deacetyltaxol is up to 0.5%. While this compound is not an intermediate in Taxol biosynthetic route, it can be converted into Taxol by de-glycosylation and acetylation. Here, we improve the catalytic efficiency of 10-deacetylbaccatin III-10-O-acetyltransferase (DBAT) of Taxus towards 10-deacetyltaxol, a de-glycosylated derivative of 7-β-xylosyl-10-deacetyltaxol to generate Taxol using mutagenesis. We generate a three-dimensional structure of DBAT and identify its active site using alanine scanning and design a double DBAT mutant (DBAT$^{G38R/F301V}$) with a catalytic efficiency approximately six times higher than that of the wild-type. We combine this mutant with a β-xylosidase to obtain an in vitro one-pot conversion of 7-β-xylosyl-10-deacetyltaxol to Taxol yielding 0.64 mg ml$^{-1}$ Taxol in 50 ml at 15 h. This approach represents a promising environmentally friendly alternative for Taxol production from an abundant analogue.

[1] State Key Laboratory of Bioactive Substance and Function of Natural Medicines, Institute of Materia Medica, Chinese Academy of Medical Sciences and Peking Union Medical College, 1 Xian Nong Tan Street, Beijing 100050, China. [2] Key Laboratory of Biosynthesis of Natural Products, National Health and Family Planning Commission of PRC, Institute of Materia Medica, Chinese Academy of Medical Sciences and Peking Union Medical College, 1 Xian Nong Tan Street, Beijing 100050, China. Correspondence and requests for materials should be addressed to P.Z. (email: zhuping@imm.ac.cn).

Taxol or paclitaxel (generic name) is one of the most well-known anticancer drugs among natural products[1]. It exerts anticancer activity through promoting tubulin assembly into microtubules and preventing their disassembly[2]. Indications include refractory ovarian cancer, breast cancer and squamous cancers[1]. Taxol is also used to treat psoriasis[3], rheumatoid arthritis[4] and restenosis[5] for which Taxol is used for the coronary stent or balloon catheter coating[6–8]. In the last decade, at least nine new taxoid derivatives or formulations have undergone clinical trials[1], including Nab-paclitaxel, Cabazitaxel and Larotaxel.

Taxol is mainly produced by yew trees. With an extremely low content, its concentration ranges from 0 to 0.069% (tree material dry weight), with the highest percentage found in the bark[9]. Therefore, at the early stage of drug development, the destructive collection of bark for Taxol threatened wild resources and was ultimately prohibited around the world[10]. For example, all *Taxus* species were under the Grade I national protection in the National Key Protected Wild Plants List (1999) published by the Chinese government (http://www.gov.cn/gongbao/content/2000/content_60072.htm). *T. wallichiana* was also listed on Appendix II of the Convention on International Trade in Endangered Species of Wild Fauna and Flora (CITES) in 1995, and afterwards other *Taxus* species, such as *T. chinensis* and *T. cuspidata*, were added to the Appendix II. Consequently, alternative Taxol sources have been explored to meet increasing clinical requirements[11–14]. Currently, Taxol produced from seedling culture, forestation of yew trees and chemical semi-synthesis of Taxol from the precursor 10-deacetylbaccatin III (10-DAB) have become the major Taxol sources for clinical supply[15,16]. 10-DAB is also one of the key intermediates in the biosynthetic pathway of Taxol and is converted into baccatin III by the enzyme 10-deacetylbaccatin III-10-*O*-acetyltransferase (DBAT)[17].

7-β-Xylosyl-10-deacetyltaxol (XDT) is an analogue of Taxol. This analogue is not involved in the Taxol biosynthetic pathway from 10-DAB to Taxol, but may share the same precursor 10-DAB. Its content in yew trees is as much as 0.5% (dry weight of the bark)[18]. XDT content is even higher than that of Taxol in nursery cultivated yew trees[19,20]. This analogue is usually discarded during Taxol extraction, causing both resource waste and potential environmental pollution. If the 7-β-xylosyl group is removed by a specific β-xylosidase, then the resulting 10-deacetyltaxol (DT) can also be used for the chemical semi-synthesis of Taxol by acetylation at the C10 position[21,22]. The corresponding bacteria with the β-xylosidase activity were explored by several other groups[23–27], in which a nanoscale cellulosome-like multienzyme complex was found from *Cellusimicrobium cellulans*[27]. Recently, we even cloned and characterized a new β-xylosidase (designated as LXYL-P1-2) from *Lentinula edodes*, which showed much lower sequence identities than other known enzymes including the reported β-xylosidases[28,29]. This enzyme can specifically hydrolyse the β-xylosyl group of XDT and is heterologously expressed in *Pichia pastoris*. We also achieved a large-scale fermentation (up to 1,000-l fermenter)[30,31] and bioconversion (up to 10-l reaction)[30,32] of the recombinant yeast, and the DT yield reached $10.58 \, \mathrm{g \, l^{-1}}$ in one-liter reaction volume, which was much higher than the earlier reports[23–26] (Supplementary Table 1).

These efforts prompted the exploration of an environmentally friendly method to acetylate the C10 position of DT by acetyltransferase, and to even combine the enzyme with LXYL-P1-2 in a one-pot reaction to produce Taxol from XDT. Apparently, DBAT is an excellent candidate, because its natural substrate 10-DAB harbours the same taxane nucleus as that of DT except for their difference at the C13 side-chain. DBAT has

been widely studied by several groups[33–36]. Menhard and Zenk[33] purified the *O*-acetyltransferase from cell suspension cultures of *T. chinensis* and characterized it. The purified acetyltransferase was found to be a monomeric protein of 71 kDa with an optimum pH of 9 and an optimum temperature of 35 °C. Walker and Croteau[34] cloned the DBAT gene from *T. cuspidata*, heterologously expressed it in *Escherichia coli* and characterized the recombinant enzyme. The recombinant enzyme was also monomeric but with a molecular weight of 49 kDa and an optimum pH of 7.4. It seemed to acetylate the 10-hydroxyl group with a high degree of regioselectivity, as the enzyme did not acetylate the 1β-, 7β- or 13α-hydroxyl groups of 10-DAB, nor did it acetylate the 5α-hydroxyl group of taxa-4(20),11(12)-dien-5α-ol[35]. Ondari and Walker[35] also demonstrated that the recombinant DBAT can acetylate docetaxel, a semi-synthetic taxoid of Taxol, at its natural C10 hydroxyl position, and suggested that acetylation at the C10 hydroxyl can occur after attachment of the phenylpropanoyl side chain to 10-DAB on the biosynthetic pathway. Pennington *et al.*[36] tried to find the acetylation activity of a partially purified DBAT from the leaves and 3-year-old cell suspension cultures of *T. cuspidata* both on 10-DAB and on DT. They observed the formation of baccatin III from 10-DAB in the presence of acetyl CoA, while repeated attempts to detect significant DT acetyltransferase activity were not successful. The inconsistent appearance of Taxol in the assays was thought to be the possible presence of an uncharacterized 10-*O*-acetyl-transferase as a contaminant in the enzyme preparation. Moreover, all of the products were not subjected to spectroscopic elucidation.

Here, we confirm the enzymatic acetylation of DT at its C10 hydroxyl position via the recombinant DBAT of *T. cuspidata* by spectroscopic detection. However, its catalytic efficiency was much lower than that on the natural substrate 10-DAB. Thus, we use protein engineering of DBAT to improve its enzymatic activity against DT, generating a double mutant DBAT[G38R/F301V] that has a $k_{cat}/K_m$ of $3.54 \, \mathrm{M^{-1} \, s^{-1}}$ on DT (DBAT with a $k_{cat}/K_m$ of $0.61 \, \mathrm{M^{-1} \, s^{-1}}$). Finally, we construct an *in vitro* one-pot reaction system harbouring LXYL-P1-2, and the improved acetyltransferase to enzymatically convert XDT into Taxol (Fig. 1). This approach might be a promising alternative for Taxol production.

## Results

**Activities of the recombinant DBATs against 10-DAB and DT**. Six full-length DBAT cDNAs derived from *Taxus cuspidata* (GenBank accession: Q9M6E2.1), *T. brevifolia* (GenBank accession: EU107143.1), *T. baccata* (GenBank accession: AF456342.1), *T. canadensis* (GenBank accession: EU107134.1), *T. wallichiana var* (GenBank accession: EU107140.1) and *T. x media* (GenBank accession: AY452666.1) were synthesized and expressed in *E. coli*. The purified recombinant DBATs were obtained through Ni-affinity chromatography and molecular size exclusion chromatography, with purities greater than 95%. The catalytic activities of the DBATs against 10-DAB (control) and DT were detected, as shown in Fig. 2. Except for the recombinant DBAT of *T. baccata* origin whose activity was nearly undetectable, all other recombinant enzymes exhibited acetylation activity against 10-DAB and DT, in which the DBAT of *T. canadensis* exhibited the lowest conversion rate (Fig. 2). The optimum temperatures for 10-DAB and DT were 40 °C (Fig. 2a) and 37.5 °C (Fig. 2c), respectively. While the optimum pH values for both of the substrates were 5.5 (Fig. 2b,d), which was not consistent with the result in an earlier report on DBAT[34]. Under the same conditions, the DBATs of *T. wallichiana var.*, *T. brevifolia*, *T. cuspidata* and *T. x media* origins showed similar conversion rates for 10-DAB (Fig. 2a,b),

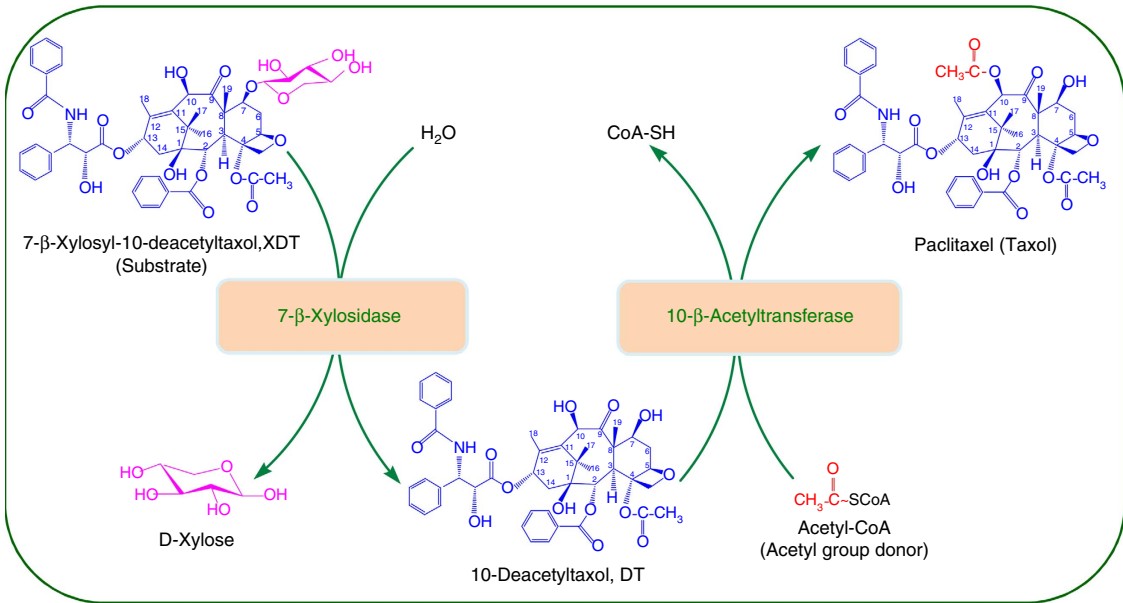

**Figure 1 | One-pot reaction system for the bioconversion of XDT to Taxol.** The *in vitro* one-pot reaction system contained a specific 7-β-xlyosidase, the improved 10-β-acetyltransferase, the substrate 7-β-xylosyl-10-deacetyltaxol (XDT) and the acetyl group donor acetyl-CoA, respectively. The 7-β-xylosidase deglycosylates XDT into the intermediate product 10-deacetyltaxol (DT) and then 10-β-acetyltransferase acetylates 10-deacetyltaxol into the final product Taxol.

while the DBATs of *T. brevifolia* and *T. cuspidata* were obviously higher than those of *T. x media* and *T. wallichiana var*. regarding conversion rates against DT (Fig. 2c,d). Accordingly, under optimum conditions and among the six recombinant enzymes, the DBATs of *T. cuspidata*, *T. brevifolia*, *T. canadensis* and *T. wallichiana var.* showed similar high specific activities (from 198.7 to 215.7 U mg$^{-1}$) against 10-DAB, almost 200 times higher than that of *T. baccata* (1.1 U mg$^{-1}$) (Fig. 2e, Supplementary Table 2) which may partially explain why the latter accumulated significantly more 10-deacetylbaccatin III in the twigs of *T. baccata* than other species[37]. Regarding the substrate DT, the DBATs of *T. cuspidata* and *T. brevifolia* showed the highest specific activities ($\sim$0.26 U mg$^{-1}$), again over 80 times more than that of *T. baccata* (0.003 U mg$^{-1}$) (Fig. 2f; Supplementary Table 2). However, all of the specific activities against DT dropped nearly three orders of magnitude compared to those of the same enzymes against the natural substrate 10-DAB (Fig. 2e,f; Supplementary Table 2).

The catalytic products of the recombinant DBAT of *T. cuspidata* were chosen for high-performance liquid chromatography/mass spectrometry (HPLC-MS) analysis. The $[M+H]^+$ and $[M+Na]^+$ peaks of the product in DBAT + DT group were 854.21 and 876.46, respectively (Fig. 3i), which were consistent with those of the standard ($m/z$ 853.75 $[M+H]^+$, 876.07 $[M+Na]^+$) (Fig. 3g) and previous literature[38]. Furthermore, the nuclear magnetic resonance (NMR) spectroscopy data (Supplementary Fig. 1, Supplementary Table 3) of the purified product were in accordance with those reported in additional literature[39]. Thus, the acetylated DT was demonstrated to be Taxol.

**Three-dimensional structure prediction of DBAT.** To improve the specific activity of DBAT against the unnatural substrate DT, we carried out site-directed mutagenesis of the protein via a 'semi-rational design' strategy. Because its three-dimensional (3D) structure was not currently available, DBAT structure and active centre

were predicted by molecular modelling through online prediction software, Swissmodel (http://swissmodel.Expasy.org/). Other tools included Discovery Studio 4.1, Procheck, Autodock 4.4, Autodock Vina, Chimera and Chemical Office. DBAT belongs to a plant acyl-CoA dependent acyltransferase superfamily, BAHD[40], and is grouped into clade V in which 3D structures of *Coffea canephora* HCT: 4G22, *Sorghum bicolor* HCT: 4KE4 and *Panicum virgatum* PvHCT2: 5FAL have been reported. Although the sequences of the BAHD superfamily members are divergent, evidence has shown that these proteins possess highly similar 3D structures[41]. On the basis of the sequence alignment between DBAT and several other BAHD members (Supplementary Fig. 2), the *C. canephora* HCT (GenBank accession: ABO47805.1) was selected as the template for homology modelling of the DBAT structure. The sequence alignment between DBAT and the template is shown in Fig. 4a. The two enzymes exhibited 30% sequence identity and 45% sequence similarity, which was most similar among the aligned BAHD members. The predicted model was assessed by the Ramachandran plot[42], indicating that the model was reasonable with 87.5% of the structure in the most favored regions, and only 0.8% of the residues in the disallowed regions (Supplementary Fig. 3). The predicted DBAT structure consisted of two nearly equal domains connected through a loop and was totally composed of 14 β-sheets and 13 α-helices (Fig. 4b). DBAT showed similar hydrophobicity and distribution of electronegativity residues with the template (Supplementary Fig. 4).

It is generally accepted that the highly conserved HXXXD motif is crucial for the activity of BAHD members[43]. The motif has also been found in the DBAT structure, in which His[162] most likely acts as a catalytic site (Fig. 4b). In addition, the structure of the solvent channel of BAHD members is also critical for activity. The acyl donor substrate binding pocket of BAHD members is at the front side of the molecule (the position of the conserved DFGWG motif has been defined at the front side)[43], which helps position the acceptor substrate pocket of DBAT at the back side (Fig. 4b, Supplementary Fig. 5). To identify the putative residues participating in the binding and/or catalysis of 10-DAB or DT,

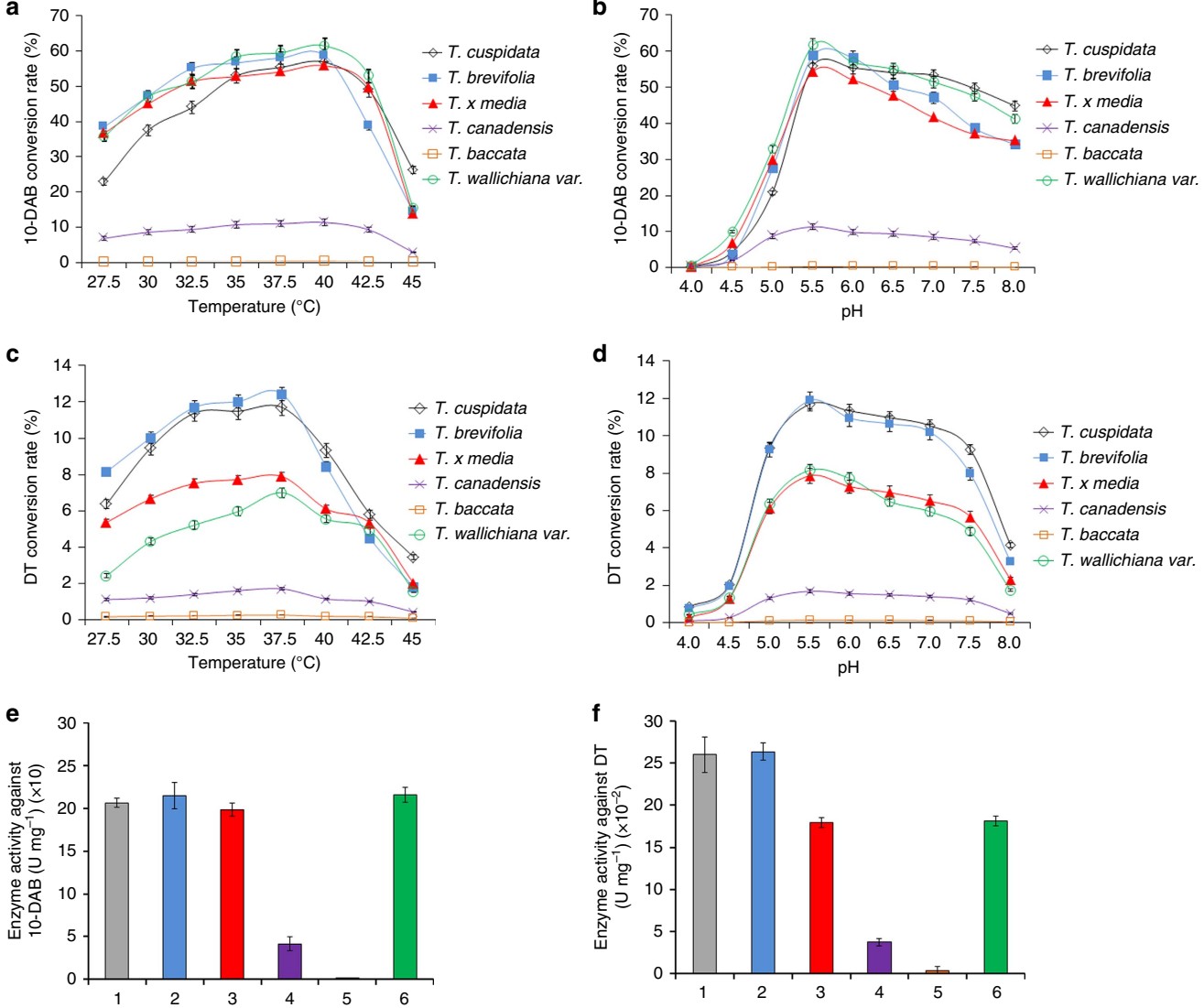

**Figure 2 | Optimum temperature and pH and the specific activities of the DBATs. (a,b)** The optimum temperature and pH of the recombinant DBATs against 10-DAB. **(c,d)** The optimum temperature and pH of the recombinant DBATs against DT. **(e)** The specific activities of the six recombinant DBATs against 10-DAB. **(f)** The specific activities of the six recombinant DBATs against DT. (1. *Taxus cuspidata*, 2. *Taxus brevifolia*, 3. *Taxus x media*, 4. *Taxus canadensis*, 5. *Taxus baccata*, 6. *Taxus wallichiana var.*). The data represent the means ± s.d., $n = 3$.

molecular docking was conducted (Fig. 4c,d). The residues within 5 Å distance from DT were selected as 'hot spots' for functional analysis.

**L-alanine scanning mutagenesis of defined amino acids.** Enzyme activity generally declines if the amino acids involved in the substrate binding or catalytic sites are subjected to L-alanine scanning mutagenesis (mutated to Ala). However, this activity may be maintained or even increased if other amino acid residues are mutated to L-alanine[44]. The following amino acid residues within the 5 Å zone around the substrate (Fig. 4d) were chosen for L-alanine scanning mutagenesis: Pro[37], Gly[38], Arg[40], Glu[41], Phe[44], Phe[160], His[162], Ile[164], Cys[165], Phe[301], Ser[351], Asn[353], Gly[359], Gly[361], Arg[363], Ser[396] and Phe[400]. Results (Fig. 5) showed that H162A had no measurable enzyme activity against 10-DAB or DT, which was consistent with the function of this residue as a potential catalytic base. Similarly, each of the R363A, G361A and I164A mutations lost virtually all activity against 10-DAB or DT, which also implied that these residues are potential substrate binding or catalytic sites. F44A activity was nearly undetectable

against DT and significantly declined against 10-DAB, which means that this residue may be involved in catalysis especially to the unnatural substrate DT. Other mutations, such as G359A, F400A, P37A and F160A, also negatively affected enzyme activities, but the impact of each mutation on 10-DAB and DT were quite different. However, F301A and G38A activities against DT significantly increased compared with that of the wild-type control, leading to 1.6 (F301A) and 1.45 (G38A) times more active than the wild-type, respectively. But their activities against 10-DAB were apparently degraded (F301A) or slightly decreased (G38A) (Fig. 5, Supplementary Table 4).

**Saturation mutagenesis of Gly[38] and Phe[301].** To identify more active mutants against DT, we performed saturation mutagenesis of Gly[38] and Phe[301] using degenerated primers and the whole plasmid PCR amplification technique, changing the codon to NNK (N = A/T/C/G, K = G/T). A total of 38 mutants from Gly[38] and Phe[301] were obtained and activities against DT were detected as shown in Fig. 6a,b (also see Supplementary Table 5 and 6). The mutants DBAT[G38R], DBAT[G38S], DBAT[G38D], DBAT[G38H],

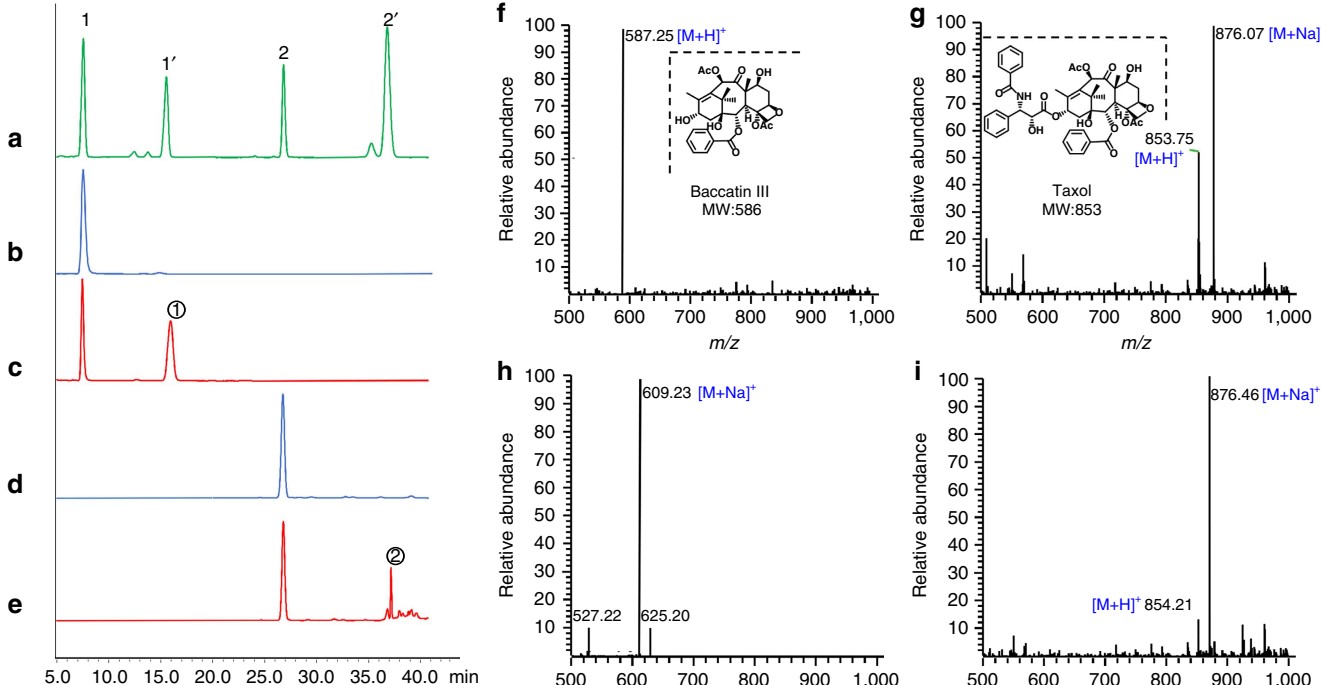

**Figure 3 | HPLC-MS analysis of the DBAT acetylated products.** HPLC analysis: (**a**) Standards of 10-DAB (1), baccatin III (1′), DT (2) and Taxol (2′). (**b**) Control of (**c**) (only 10-DAB, without the enzyme). (**c**) Reaction of DBAT on 10-DAB. (**d**) Control of (**e**) (only DT, without the enzyme). (**e**) Reaction of DBAT on DT. ESI-MS analysis: (**f,g**) Standards of baccatin III (1′) and Taxol (2′) in **a**. (**h,i**) Products ① and ② in **c,e**.

DBAT[F301V] and DBAT[F301C] surpassed the corresponding DBAT[G38A] or DBAT[F301A] in terms of activity, for which DBAT[G38R] and DBAT[F301V] were 2.19 and 2.85 times more active than the wild-type, respectively.

Two mutants, DBAT[G38R] and DBAT[F301V], were combined to form the double mutant DBAT[G38R/F301V]. This combination did not change the optimum temperature and optimum pH when compared to those of the wild-type DBAT (Fig. 6c,d), but further increased enzyme activity 3.7 times against DT (Table 1). However, except for DBAT[G38R], which showed two times greater activity against 10-DAB, both DBAT[G38R/F301V] and DBAT[F301V] exhibited decreased activities against 10-DAB compared with the control (Table 1). Enzyme kinetic analysis showed that catalytic efficiency ($k_{cat}/K_m$) of DBAT[G38R/F301V] against DT surpassed those of DBAT[G38R] and DBAT[F301V], and was significantly higher than that of the wild-type control ($P < 0.001$, Table 2).

**Selection of DBAT[G38R/F301V] concentration in reaction system.** The substrate DT concentration was assigned to 2 mM and dissolved in the reaction buffer solution with a final concentration of 5% dimethyl sulfoxide (DMSO, v/v). The suitable enzyme concentration was determined under 37.5 °C and pH 5.5. The Taxol yield increased with the increasing amount of DBAT[G38R/F301V]. However, when the enzyme concentration reached 2.0 mg ml$^{-1}$, the increasing amplitude of Taxol yield began to plateau (Fig. 7a). Because Taxol yields were similar between enzyme concentrations of 1.5 and 2.0 mg ml$^{-1}$ at the same time points (Fig. 7a), to limit the enzyme consumption, a 1.5 mg ml$^{-1}$ enzyme concentration was used to observe changes in the concentrations of DT and Taxol in the reaction system. The Taxol yield reached 452 μg ml$^{-1}$ after 12 h (Fig. 7c). As the linear portion of the time course plot was within 3 h (Fig. 7a), the supplementing strategy was utilized beginning at 3 h (near the inflection point), and additional DBAT[G38R/F301V] was repeatedly supplemented in the amount of 1.5 mg ml$^{-1}$ in 3 h intervals (Fig. 7d). The Taxol yield reached 641 μg ml$^{-1}$ after 15 h (Fig. 7d).

**Selection of LXYL-P1-2 concentration in reaction system.** The optimum temperature and optimum pH of LXYL-P1-2 were 45 °C and 4.5, respectively[28]. Since catalytic efficiency of LXYL-P1-2 against XDT was much higher than that of DBAT[G38R/F301V] against DT ($9.7 \times 10^3$ M$^{-1}$s$^{-1}$ versus 3.5 M$^{-1}$s$^{-1}$), the following one-pot reaction used optimum conditions of DBAT[G38R/F301V], that is, 37.5 °C and pH 5.5, to maximize Taxol production. The final concentration of XDT assigned was 2 mM, and the influence of LXYL-P1-2 concentration on DT yield was detected (Fig. 7b). DT yields reached 830, 987, 1,054 and 1,093 μg ml$^{-1}$ at 12 h, when LXYL-P1-2 concentrations were 0.25, 0.5, 1.0 and 1.5 mg ml$^{-1}$, respectively. All of the yields met the requirement for the next reaction from DT to Taxol and the relatively lower concentration of 0.5 mg ml$^{-1}$ LXYL-P1-2 was chosen in the following one-pot reaction system.

**Construction of one-pot reaction system.** In the 1-ml one-pot reaction system, the two enzymes and the substrate XDT were added and the reaction was performed under the aforementioned conditions. Taxol yields reached 657 and 349 μg ml$^{-1}$, respectively, after 15 h with or without the supplementing strategy (Fig. 7e,f).

The one-pot reaction volume was amplified from 1 ml to 50 ml with the supplementing strategy and similar yields of Taxol were obtained, in which the Taxol yield reached 635 μg ml$^{-1}$ in the 50-ml reaction volume at 15 h (Fig. 7g, Supplementary Table 7). The product from the entire 50-ml reaction was extracted through ethyl acetate. Once concentrated, the product was purified by HPLC (Fig. 7h) for structure elucidation and was confirmed to be Taxol by MS, [1]H-NMR and [13]C-NMR analysis (Supplementary Fig. 1, Supplementary Table 3).

**Discussion**

Taxol is a highly-oxygenated natural product with eleven chiral carbons. This diterpenoid compound harbours a main skeleton of A-, B- and C-rings, an oxetane ring (D-ring), the

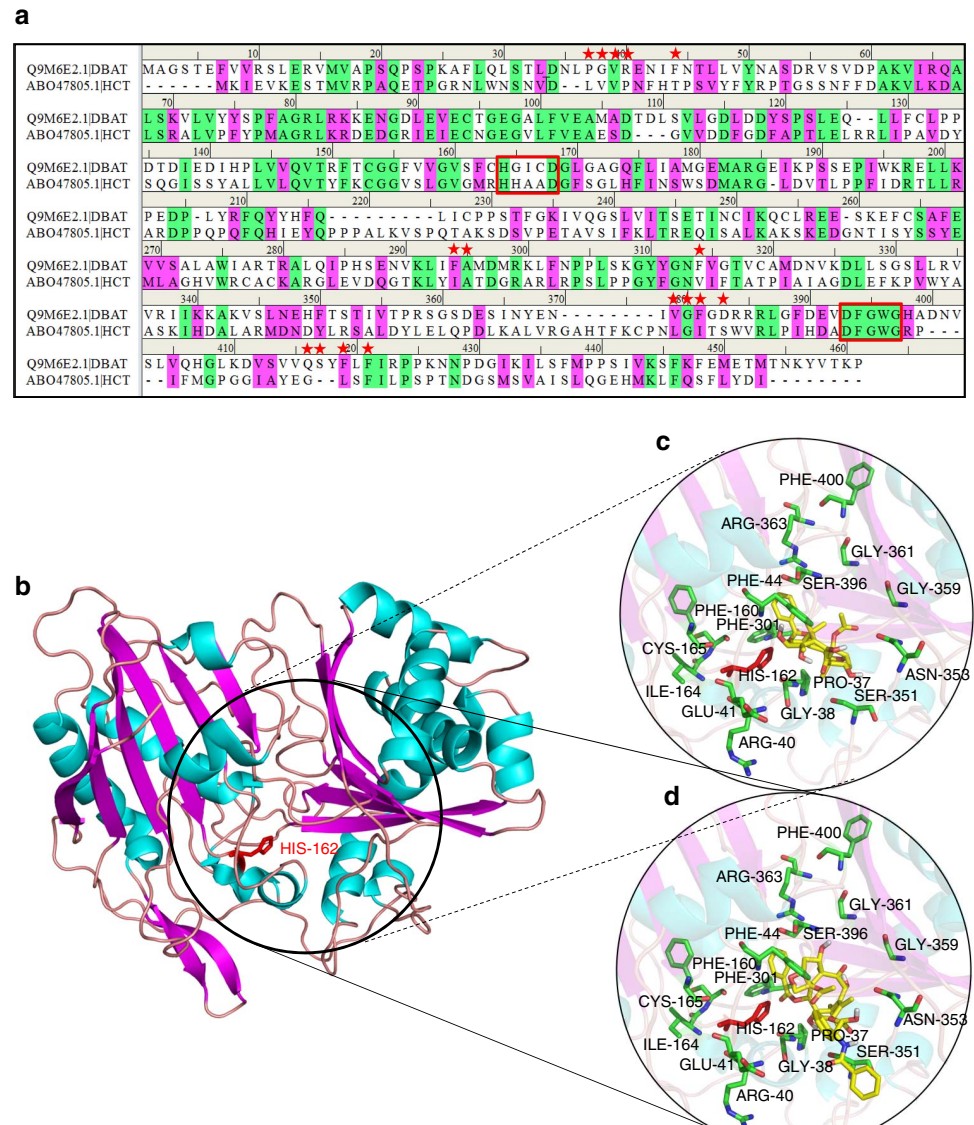

**Figure 4 | The DBAT structure predicted by homology modeling (template: HCT) and molecular docking of DBAT with the substrates.** (**a**) Sequence alignment between DBAT and the template HCT (ABO47805.1). Identical and similar residues are highlighted in green and pink, respectively. Residues predicted to be involved in acyl accepter substrate binding are highlighted with red stars, and the conserved HXXXD and DFGWG motifs are marked in red box. (**b**) Homology model of DBAT constructed with Swiss-Model (the distance 5 Å around the substrate position was circled). (**c**) Enlarged view of molecular docking of DBAT with 10-DAB. (**d**) Enlarged view of molecular docking of DBAT with DT.

N-benzoyphenylisoserine side-chain appended to C13 of the A-ring, and the benzoyl group at C2 of the B-ring. Other functional groups include two hydroxyl groups (at C1 and C7) and two acetyl groups (at C4 and C10). It has been postulated that 20 enzymes are involved in 19 steps of the biosynthetic pathway from the universal diterpenoid precursor geranylgeranyl diphosphate (GGPP) to Taxol, of which six corresponding genes remains to be identified, i.e., C1β-hydroxylase, C4,C20-epoxydase, C4b,C20-oxomutase, C9α-hydroxylase, C9-oxidase and C2′ side-chain hydroxylase[45–47]. The first committed step is the cyclization of GGPP to taxa-4(5), 11(12)-diene. This olefin is then structurally modified by eight cytochrome P450-mediated oxygenations, two subsequent acetylations, a benzoylation, oxetane ring formation and oxidation at C9 en route to the late intermediate baccatin III, to which the C13 side-chain is attached to yield the final product Taxol[45–49]. In comparison with Taxol, 7-β-xylosyl-10-deacetyltaxol (XDT) possesses a β-xylosyl group at the C7 position but lacks an acetyl group at the C10 position.

XDT and Taxol most likely share the same precursor 10-DAB, but the former is synthesized by an unknown branch pathway most likely through the intermediate DT (the compound is also frequently isolated from *Taxus* species in the similar amount of Taxol[37]). The abundant production of XDT is likely a kind of detoxifying mechanism to the plant host, since as the mitotic inhibitor, the *in vitro* cytotoxicity of Taxol was much higher than that of XDT (Supplementary Table 8). Baccatin III is one of the key intermediates in the biosynthetic pathway of Taxol, which is produced by the C10 hydroxyl acetylation of 10-DAB catalysed by DBAT. This enzyme has been systematically investigated by other groups[33–36]. The present study revealed that the DBATs of different *Taxus* species exhibited various activities on 10-DAB. Among the six studied DBATs, higher and similar activities against 10-DAB were found in *T. cuspidata*, *T. brevifolia*, *T. x media* and *T. wallichiana var.* origins, but the activities of *T. canadensis* and *T. baccata* origins were obviously lower than those of the four species, with the activity of *T. baccata* origin

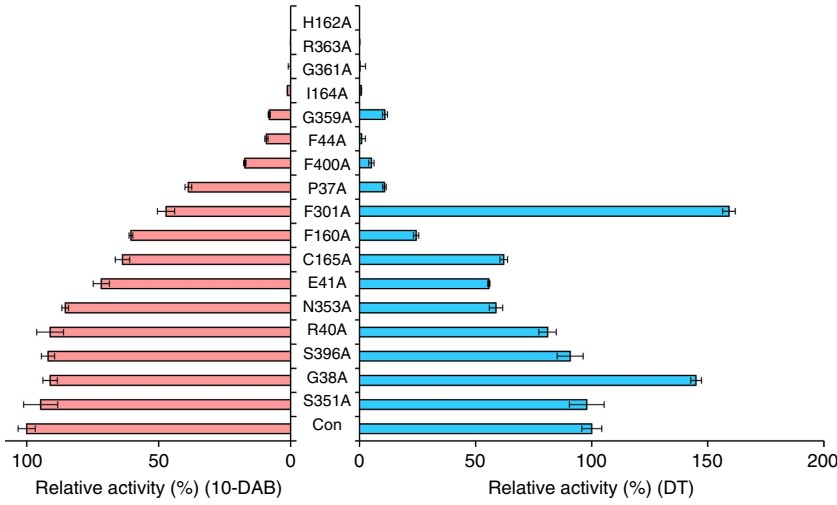

**Figure 5 | Relative activities of ʟ-alanine scanning mutations of DBAT against 10-DAB (left) and DT (right).** Con: DBAT. The data represent the means ± s.d., $n = 3$.

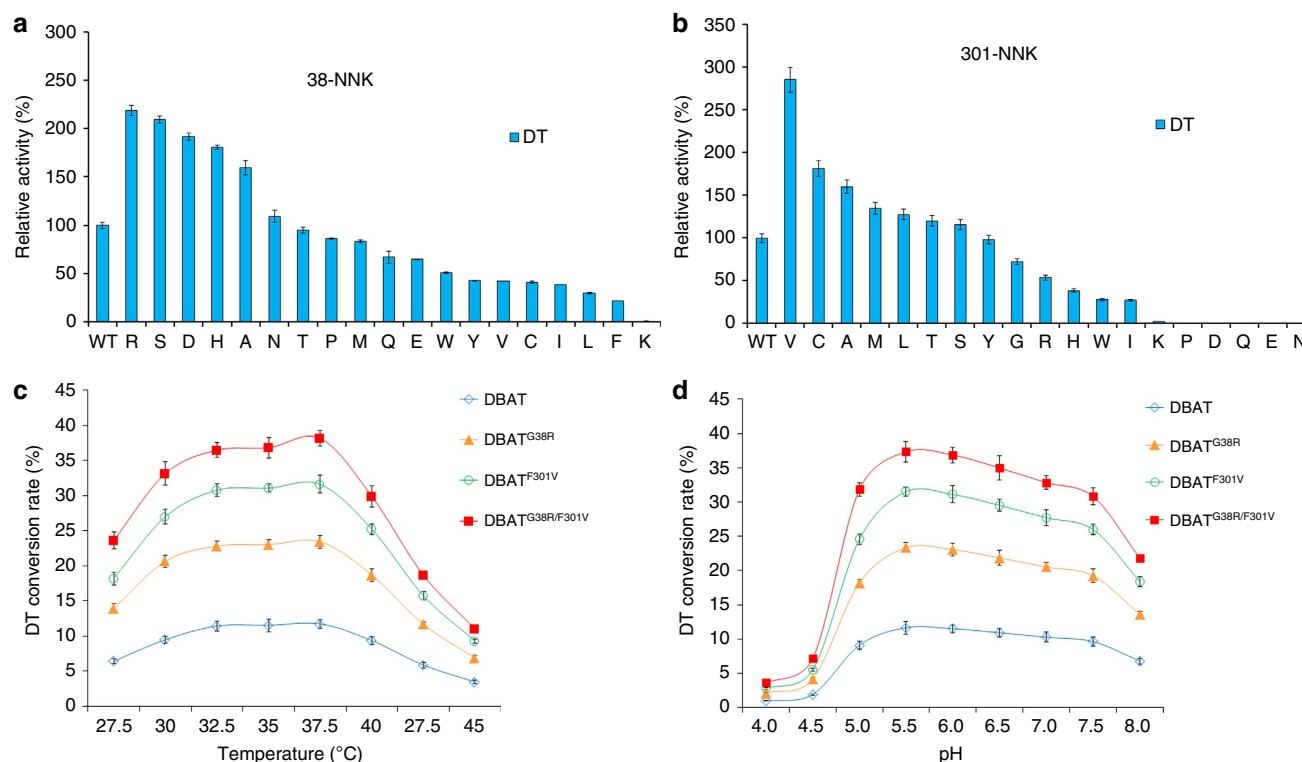

**Figure 6 | Activities of Gly[38] and Phe[301] saturation mutagenesis on DT and temperature- and pH- course profiles of the mutants.** (**a**) Relative activities of Gly[38] saturation mutations on DT. (**b**) Relative activities of Phe[301] saturation mutations on DT. (**c**) Temperature-course profiles of the DBAT mutants. (**d**) pH-course profiles of the DBAT mutants. The data represent the means ± s.d., $n = 3$.

**Table 1 | Specific activities of the DBAT mutants against DT and 10-DAB.**

| | DT | | 10-DAB | |
|---|---|---|---|---|
| | U mg$^{-1}$ ($\times 10^{-2}$) | Relative activity (%) | U mg$^{-1}$ ($\times 10$) | Relative activity (%) |
| DBAT | 26.00 (± 1.12) | 100.00 | 20.66 (± 0.67) | 100.00 |
| DBAT$^{G38R}$ | 56.88 (± 3.11)** | 218.77 | 44.34 (± 3.76)** | 214.62 |
| DBAT$^{F301V}$ | 74.10 (± 8.45)** | 285.00 | 10.87 (± 0.23) | 52.60 |
| DBAT$^{G38R/F301V}$ | 96.21 (± 5.13)** | 370.03 | 14.32 (± 0.71) | 69.31 |

The data represent the means ± s.d., $n = 3$. *$P < 0.05$ versus DBAT (Control), **$P < 0.01$ versus DBAT (Control) (Student's $t$-test).

**Table 2 | Kinetic parameters of the DBAT mutants against DT.**

| | DBAT | DBAT$^{G38R}$ | DBAT$^{F301V}$ | DBAT$^{G38R/F301V}$ |
|---|---|---|---|---|
| $V_{max}$ ($\mu$M min$^{-1}$) | 0.204 ($\pm$ 0.004) | 0.499 ($\pm$ 0.029)*** | 0.242 ($\pm$ 0.008)** | 0.918 ($\pm$ 0.096)*** |
| $K_m$ (mM) | 0.546 ($\pm$ 0.029) | 0.368 ($\pm$ 0.061)* | 0.178 ($\pm$ 0.017)*** | 0.424 ($\pm$ 0.133) |
| $k_{cat}$ (s$^{-1}$)($\times 10^{-3}$) | 0.334 ($\pm$ 0.001) | 0.815 ($\pm$ 0.005)*** | 0.396 ($\pm$ 0.013)** | 1.499 ($\pm$ 0.157)*** |
| $k_{cat}/K_m$ (M$^{-1}$s$^{-1}$) | 0.611 ($\pm$ 0.243) | 2.218 ($\pm$ 0.787)* | 2.22 ($\pm$ 0.771)* | 3.539 ($\pm$ 0.118)*** |

The data represent the means $\pm$ s.d., $n = 3$. *$P < 0.05$ versus DBAT (Control), **$P < 0.01$ versus DBAT (Control), ***$P < 0.001$ versus DBAT (Control) (Student's $t$-test).

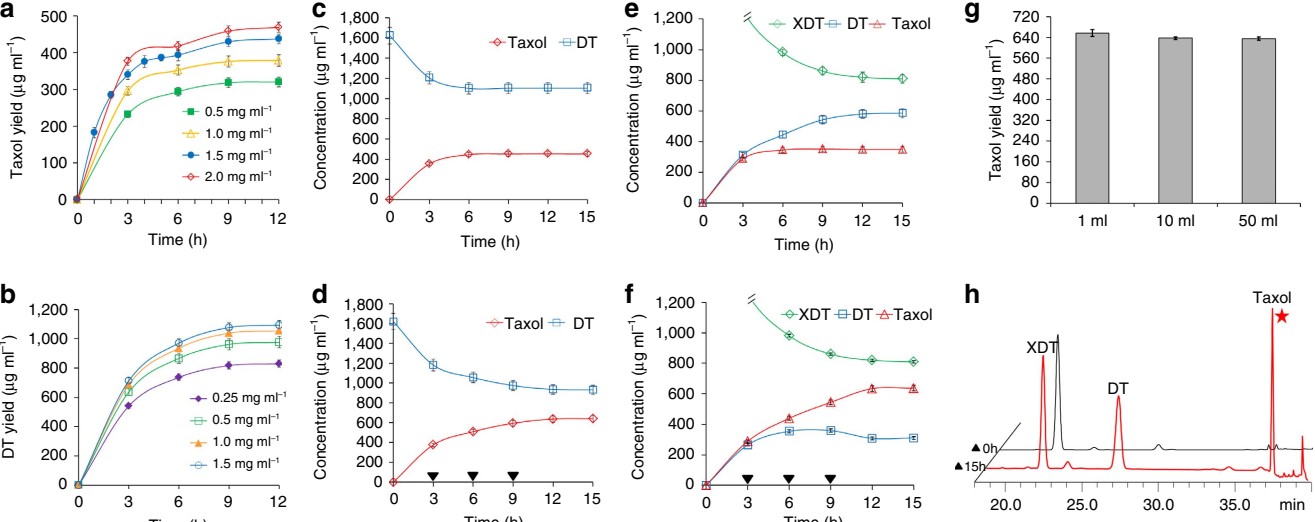

**Figure 7 | Time-concentration curves of XDT and DT as well as Taxol in the reactions and the Taxol yields in the one-pot reaction system.** (**a**) Time–concentration curves of Taxol with DT as the substrate and catalysed by different concentrations of DBAT$^{G38R/F301V}$. (**b**) Time–concentration curves of DT with XDT as the substrate and catalysed by different concentrations of LXYL-P1-2. (**c**) Time–concentration curves of DT and Taxol catalysed by 1.5 mg ml$^{-1}$ of DBAT$^{G38R/F301V}$ without DBAT$^{G38R/F301V}$ supplement. (**d**) Time–concentration curves of DT and Taxol catalysed by 1.5 mg ml$^{-1}$ of DBAT$^{G38R/F301V}$ with DBAT$^{G38R/F301V}$ supplement (downward black triangle, adding time). (**e**) Time–concentration curves of XDT, DT and Taxol in the one-pot reaction system (0.5 mg ml$^{-1}$ of LXYL-P1-2 plus 1.5 mg ml$^{-1}$ of DBAT$^{G38R/F301V}$) without DBAT$^{G38R/F301V}$ supplement. (**f**) Time–concentration curves of XDT, DT and Taxol in the one-pot reaction system with DBAT$^{G38R/F301V}$ supplement (downward black triangle, adding time). (**g**) Scaling-up of the one-pot reaction. (**h**) HPLC analysis of the one-pot reaction at 15 h (reaction volume: 50 ml). The data represent the means $\pm$ s.d., $n = 3$.

being the lowest (Fig. 2, Supplementary Table 2). The decreased activity of *T. baccata* origin against 10-DAB may at least partially explain why this species frequently accumulates significantly more 10-DAB in its twigs than other species[37,50], since apart from the variation of the expression levels, such a low activity may hamper the enzymatic step from 10-DAB to baccatin III, leading to the abundant accumulation of 10-DAB in the plant.

To the unnatural substrate DT, the enzyme activity of each species was only about 0.1% of the same enzyme against the natural substrate 10-DAB, implying that the larger C13 side-chain of DT sharply decreased the fitness of the enzyme to the unnatural substrate. This finding also suggests that under natural conditions acetylation at the C10 position is less likely to occur after attachment of the phenylpropanoyl side chain to 10-DAB on the biosynthetic pathway.

To explore the relationship between the structure and function of DBAT, especially to improve its catalytic efficiency against DT, this study modelled a 3D structure of DBAT on the structure of *C. canephora* HCT (GenBank accession: ABO47805.1), as the template (30% sequence identity and 45% sequence similarity with DBAT) and obtained a reasonable model with 87.5% of the structure in the most favoured regions. The model indicated that the enzyme is composed of two nearly equal domains connected through a loop (Fig. 4b). The highly conserved HXXXD and DFGWG motifs help position the active centre and acetyl

acceptor pocket of DBAT at the back side, in which His[162] probably acts as a catalytic site. After molecular docking, the points within 5 Å distance from DT were selected as 'hot spots' for individual mutations of the residues to L-alanine (L-alanine scanning) to identify putative residues participating in the binding and/or catalysis of 10-DAB or DT (Fig. 4c,d). The strategy of the 'hot spots' selection for semi-rational mutagenesis has been demonstrated to be helpful in determining the active sites of the enzyme and increasing likelihood of finding improved enzymes in protein engineering[44,51,52].

Results confirmed the residue His[162] as the catalytic base, as the H162A mutation almost fully lost enzyme activity. Similar phenomena were found in R363A, G361A, I164A, F44A, G359A and F400A mutations, implying that Arg[363], Gly[361], Ile[164], Phe[44], Gly[359] and Phe[400] are also potential substrate binding or catalytic sites (Fig. 5). Furthermore, Phe[44], Phe[400] and Pro[37] are more crucial to DT than to 10-DAB in enzymatic catalysis. The impact of the amino acid residues around the substrate within the 5 Å region on the acetylation of 10-DAB or DT is respectively summarized in Fig. 8, which exhibits the extent of the activity maintenance after L-alanine scanning mutagenesis.

The residues Gly[38] and Phe[301] may not have participated in the DT binding, and the G38A and F301A mutations even showed significant increase in activities against DT (Fig. 5), which led to performing site-specific saturation mutations to identify more

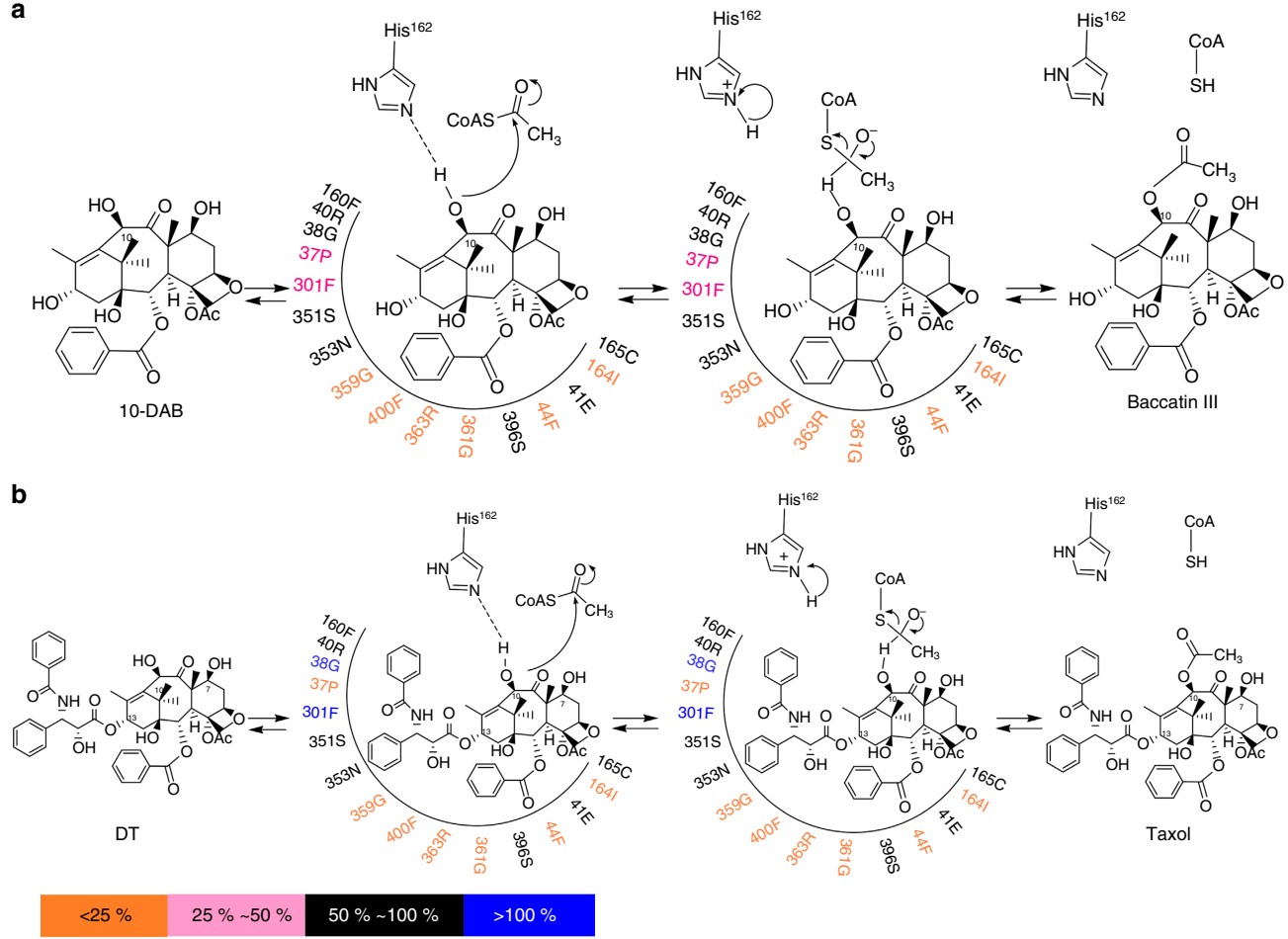

**Figure 8 | The putative catalytic processing of 10-DAB or DT by DBAT supported by DBAT activity assays after site-specific mutagenesis.** In the catalytic process, His[162] of DBAT acts as a general base catalyst and deprotonates the substrate 10-DAB or DT. (**a**) For 10-DAB, Gly[359], Phe[400], Arg[363], Gly[361], Phe[44] and Ile[164] of DBAT seem to form the substrate binding and/or catalytic sites, as supported by the significant decrease in the activity of the enzyme when these residues are mutated to L-alanine. (**b**) For DT, Pro[37], Gly[359], Phe[400], Arg[363], Gly[361], Phe[44] and Ile[164] of DBAT seem to form the substrate binding and/or catalytic sites, due to the significant decrease in the activity after L-alanine scanning mutagenesis. Gly[38] and Phe[301] do not seem to be involved in substrate binding or be part of the catalytic site, and the G38A and F301A mutations increase the activity of DBAT towards DT.

active mutants; thus, the two mutants DBAT[G38R] and DBAT[F301V] were obtained. The two mutants respectively showed 2.19 and 2.85 times more activity than the wild-type on DT (Table 1). Furthermore, a synergistic effect between the two mutations (G38R and F301V) was observed in the double mutant DBAT[G38R/F301V], which showed approximately six times the catalytic efficiency on DT compared with DBAT (Table 2).

The double mutant DBAT[G38R/F301V] was subjected to molecular docking with DT and the virtual complex was compared with DBAT-DT virtual complex in which the amino acid residues of 38 and 301 sites were highlighted (Fig. 9). As the side-chain of Arg is more hydrophilic than that of Gly, and the 38 site is near the surface of DBAT (Fig. 9a,b), the increased hydrophilic property of G38R mutation is likely one of the reasons that led to the elevated catalytic efficiency ($k_{cat}/K_m$) (Table 2). The G38R mutation also positively influenced the enzyme activity on 10-DAB (Table 1), which suggests that the mutant favours the entry of both substrates. The site Phe[301] was in the interior of the substrate solvent channel of DBAT and its phenyl side-chain may hamper the entry of DT deeper into the binding pocket. In contrast, Val not only maintains the hydrophobicity but is smaller in volume (98.71 Å$^3$) than that of Phe (131.1 Å$^3$). Thus the F301V mutation may reduce the hindrance of the enzyme against DT (Fig. 9c,d), which likely explains why this mutation led to increased catalytic efficiencies for DT (Table 2). In addition, the activity of F301V mutation towards its natural substrate 10-DAB was nearly half that of the wide type DBAT (Table 1). Altogether, these results indicate a combined favourable effect of a more opened pocket and a bulkier substrate such as DT, while 10-DAB could go too far due to its smaller size. A similar phenomenon of declined activity towards its natural substrate 10-DAB was also observed in the F301A mutation (Fig. 5). These results suggested that the bulkier Phe[301] may play a role for switching the acyl acceptor substrate specificity of DBAT. Since the possible conformational changes of the amino acid side-chains induced by the substrate and the interactions between the substrate and the related amino acid side-chains cannot be more exactly characterized by the molecular docking and the virtual 3D structure analysis, a future study on the enzyme–substrate complex crystallization and the 3D structure analysis is required, which may decipher the detailed catalytic mechanism of the enzyme on the substrate.

Fully utilizing the abundant analogue XDT for production of Taxol not only reduces resource waste and potential environmental pollution, but also alleviates the shortage for the clinical supply of the drug. Conversion of XDT through DT to Taxol by the enzymatic approach is more environmentally friendly than the chemical approach. The enzymatic deglycosylation of XDT to DT has been

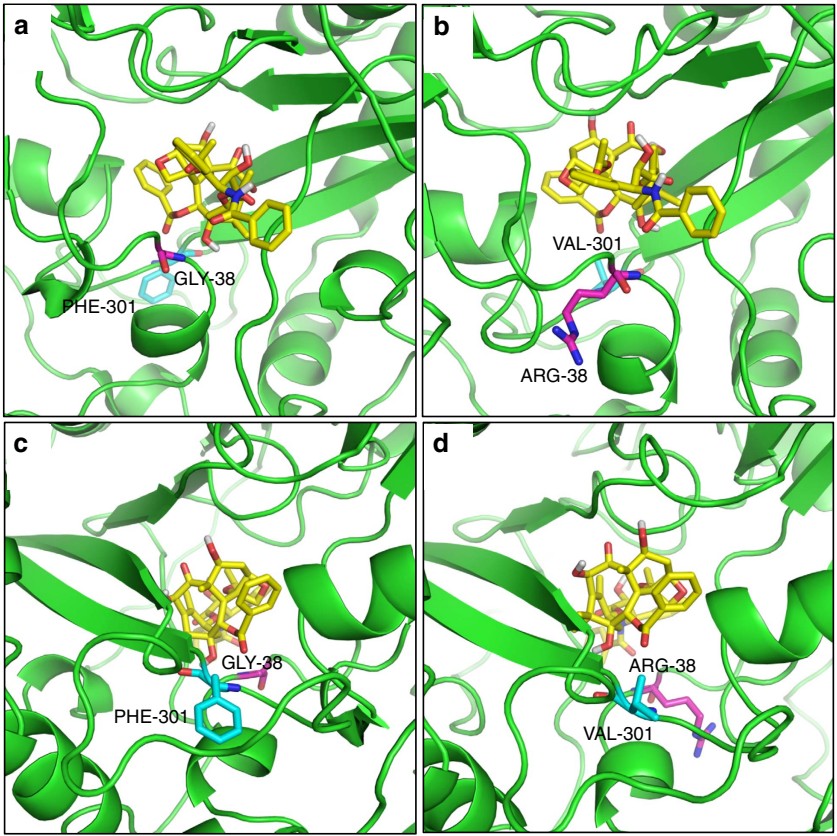

**Figure 9 | Molecular docking of DBAT and DBAT$^{G38R/F301V}$ with DT.** DT carbon atoms are coloured in yellow and oxygen atoms in red. The side chains of Gly$^{38}$/Arg$^{38}$ and Phe$^{301}$/Val$^{301}$ are also shown. Gly$^{38}$/Arg$^{38}$ carbon atoms are coloured in magenta and oxygen atoms in red. Phe$^{301}$/Val$^{301}$ carbon atoms are coloured in cyan and nitrogen atoms in blue. (**a**) Side view of DBAT with DT, showing Gly$^{38}$; (**b**) Side view of DBAT$^{G38R/F301V}$ with DT, showing Arg$^{38}$. The increased hydrophilic properties of Arg$^{38}$ compared to that of Gly$^{38}$ favors the entry of both DT and 10-DAB substrates and leads to an elevated catalytic efficiency in the G38R mutant. (**c**) Opposite side view of DBAT with DT, showing Phe$^{301}$. (**d**) Opposite side view of DBAT$^{G38R/F301V}$ with DT, Showing Val$^{301}$. Val$^{301}$ maintains the hydrophobicity but has a smaller volume (98.71 Å$^3$) than that of Phe (131.1 Å$^3$), which leads to an increased catalytic efficiency in the F301V mutant due to the reduced hindrance.

achieved by several groups$^{23–29}$, in which the specific β-xylosidase (designated as LXYL-P1-2) of the fungus *L. edodes* origin was well characterized$^{28,29}$. The enzymatic acetylation of DT to Taxol can be realized by using the 10-β-acetyltransferase double mutant DBAT$^{G38R/F301V}$, which showed the elevated activity against DT.

In the present study, the *in vitro* one-pot reaction system of de-glycosylation/acetylation was constructed with the specific β-xylosidase LXYL-P1-2 and the 10-β-acetyltransferase double mutant DBAT$^{G38R/F301V}$, using XDT as the substrate for the synthesis of Taxol. As catalytic efficiency of LXYL-P1-2 on XDT was apparently higher than on 7-xylosyltaxol, likewise, catalytic efficiency of DBAT$^{G38R/F301V}$ on DT was much higher than on XDT, thus, the reaction order should be XDT through DT to Taxol in the reaction system (Fig. 1). Additionally, since the catalytic efficiency of LXYL-P1-2 against XDT was obviously higher than that of DBAT$^{G38R/F301V}$ against DT, optimum conditions of DBAT$^{G38R/F301V}$ and DBAT$^{G38R/F301V}$ supplementing strategy were used in the one-pot reaction (2 mM of XDT, 0.5 mg ml$^{-1}$ of LXYL-P1-2, 1.5 mg ml$^{-1}$ of DBAT$^{G38R/F301V}$ at 37.5 °C and pH 5.5; additional 1.5 mg ml$^{-1}$ of DBAT$^{G38R/F301V}$ was repeatedly supplemented in 3-h intervals) to maximize the production of Taxol. The Taxol yield reached 657 µg ml$^{-1}$ at 15 h in the 1-ml reaction system (Fig. 7f), which was comparable to Taxol production (641 µg ml$^{-1}$) in the single enzyme (DBAT$^{G38R/F301V}$) reaction system that applied DT as the substrate (Fig. 7d). The two-enzyme coupled one-pot reaction was scaled up from 1 ml to 50 ml, and similar yields were

obtained, of which 0.64 mg ml$^{-1}$ Taxol was produced in 50-ml reaction (Fig. 7g, Supplementary Table 7).

As clinical demand increases, the long-term supply of Taxol will continue to be an important issue. This study may provide another choice for Taxol production from the ample analogue XDT. As activity of DBAT$^{G38R/F301V}$ against DT is not high enough, future work will be focused on the improvement of DBAT by protein engineering to elevate enzyme activity and/or stability. The supply of acetyl CoA to the reaction system is another concern, as it is not a bulk commodity and has become a limiting factor for the large-scale reaction. The practical method for massively producing acetyl CoA is probably the genetic engineering method. The *in vivo* synthetic biology (constructing biosynthetic pathway from XDT to Taxol in a microbial chassis cell) is a promising strategy to produce Taxol from the substrate XDT. In the long term, the *in vivo* and *de novo* synthetic biology for Taxol production through a microbial cell factory should be a favourable approach that does not depend on plant-based processes$^{14,53}$. The latter is based on a thorough understanding of the complex biosynthetic pathway and regulation of Taxol biosynthesis. Considering the complicated structure and certain undefined steps, obtaining the final product Taxol through such strategy is still confronted with many challenges.

## Methods

**Materials and strains.** The *dbat* gene of *Taxcus cuspidata* (GenBank accession: Q9M6E2.1) was synthesized by Viewsolid Biotech Crop (Beijing, China). Other

*dbat* genes of *Taxus brevifolia* (GenBank accession: EU107143.1), *T. baccata* (GenBank accession: AF456342.1), *T. canadensis* (GenBank accession: EU107134.1), *T. wallichiana var* (GenBank accession: EU107140.1), *and T. x media* (GenBank accession: AY452666.1) were synthesized by SynBio Research Platform at Tianjin University (Tianjin, China). Phusion polymerase, restriction enzymes, and T4 ligase were purchased from New England Biolabs (Ipswich, MA). *Escherichia coli* JM109 competent cells and plasmids were purchased from TransGen Biotech (Beijing, China). Taxol was kindly provided by Beijing Union Pharmaceutical Factory, 10-DAB and baccatin III were purchased from J&K Scientific Ltd (Beijing, China). XDT and DT (HPLC purity > 98%) were purified by our laboratory. Acetyl-coenzyme A was purchased from Sigma-Aldrich (St. Louis, MO, USA). All other chemicals were of analytical grade unless otherwise indicated.

**Recombinant protein expression and purification.** The full-length *dbat* genes were amplified using a forward primer (F: 5′-CTTAGGAGGTCA-TATGCATCATCATCATCATCATCATCATGC-3′) with *Nde* I site (underlined) and His-tag coding sequence and a reverse primer (R: GCAGGTCGACTCTA-GACTAAGGGTTTAGTTAC) with *Xba* I site (underlined). Each of the PCR products was digested by *Nde* I and *Xba* I for directional ligation into vector pCWori. After ligation, the construct was transformed into *E. coli* JM109 competent cells. *E. coli* cells harbouring the recombinant plasmids were grown overnight at 37 °C and 200 r.p.m. in 10 ml Luria-Bertani (LB) medium containing ampicillin (100 µg ml$^{-1}$) in a shake flask. The overnight cultures were suspended in 100 ml fresh Terrific Broth (TB) medium [1.2% (w/v) Tryptone, 2.4% (w/v) Yeast Extract, 0.4% (v/v) Glycerol, 17 mM KH$_2$PO$_4$, 72 mM K$_2$HPO$_4$] at a final concentration of 1% (v/v), and grown at 37 °C and 200 r.p.m. for 2~3 h until OD$_{600}$ reached 0.8. Then, isopropyl-β-D-thiogalactopyranoside (IPTG) was added at a final concentration of 1 mM and the cell cultures were incubated for an additional 18 h at 20 °C and 200 r.p.m.. Ampicillin at 100 µg ml$^{-1}$ was added to the medium when required.

The recombinant DBAT was partially purified by affinity chromatography in Ni-NTA agarose (General Electric Company, USA) according to the manufacturer's instructions. Briefly, the induced cells were resuspended in the sample buffer (20 mM imidazole, 100 mM NaCl in 20 mM Tris-HCl buffer, pH 7.5) with a concentration of 15% (v/v). After high pressure crushing (800 bar, 3 times), the sample was centrifuged with 13,400 g at 4 °C for 30 min and the supernatant was filtered through a 0.45 µm membrane for the further purification. The Ni-NTA agarose was equilibrated by the equilibration buffer (same as sample buffer). The elution buffer contained the same components in addition to 200 mM imidazole. The partially purified sample was concentrated via ultrafiltration (30 kDa, Millipore, Billerica, MA). DBAT was further purified by molecular size chromatography (Agilent Zorbax Bio Series GF-250). The equilibration buffer contained 0.1 M NaCl, 0.1 M K$_2$HSO$_4$, and 0.1 M KH$_2$SO$_4$, pH 7.5. Proteins were eluted with the same buffer at a flow rate of 0.2 ml min$^{-1}$. The purified protein was detected by SDS-PAGE gels (12%) followed by Coomassie Blue or silver staining. The concentration of protein was determined with Pierce BCA Protein Assay Kit (Thermo Fisher Scientific).

**Characterization and kinetics of the recombinant DBATs.** Determination of the optimal temperature on 10-DAB: A total reaction volume of 100 µl contained 500 µM 10-DAB, 500 µM acetyl-CoA, and 0.02 mg ml$^{-1}$ DBAT in 50 mM sodium acetate buffer, pH 5.5. The reaction was performed under the following temperatures: 27.5, 30, 32.5, 35, 37.5, 40, 42.5, and 45 °C for 20 min.

Determination of the optimal temperature on DT: A total reaction volume of 100 µl contained 500 µM DT, 500 µM acetyl-CoA, and 0.5 mg ml$^{-1}$ DBAT in 50 mM sodium acetate buffer, pH 5.5. The reaction was performed under the following temperatures: 27.5, 30, 32.5, 35, 37.5, 40, 42.5, and 45 °C for 180 min.

Determination of the optimal pH on 10-DAB: A total reaction volume of 100 µl contained 500 µM 10-DAB, 500 µM acetyl-CoA, and 0.02 mg ml$^{-1}$ DBAT in 50 mM sodium acetate buffer with pH 4.0, 4.5, 5.0, 5.5 and 6.0; or in 50 mM sodium phosphate buffer with pH 6.5, 7.0, 7.5 and 8.0. The reaction was performed under 40 °C for 20 min.

Determination of the optimal pH on DT: A total reaction volume of 100 µl contained 500 µM DT, 500 µM acetyl-CoA, and 0.5 mg ml$^{-1}$ DBAT in 50 mM sodium acetate buffer with pH 4.0, 4.5, 5.0, 5.5, and 6.0; or in 50 mM sodium phosphate buffer with pH 6.5, 7.0, 7.5 and 8.0. The reaction was performed under 40 °C for 180 min.

Reactions were terminated by adding 500 µl methanol and analysed by HPLC [Cosmosil-C18, flow rate: 1 ml min$^{-1}$, column temperature: 28 °C, UV detection wavelength: 230 nm, mobile phase: acetonitrile (A)–water (B). Gradient elution conditions see Supplementary Tables 9 and 10]. The conversion rate was calculated as the product peak area/(substrate peak area + product peak area) × 100%.

Enzymatic activity analysis: The enzymatic activities of DBATs were determined by measuring the amount of baccatin III (substrate 10-DAB) or Taxol (substrate DT) under the optimal conditions. To measure baccatin III production, 100 µl reaction volume contained 500 µM 10-DAB, 500 µM acetyl-coenzyme A, and 0.02 mg ml$^{-1}$ DBAT in 50 mM sodium acetate buffer, pH 5.5. The reaction was performed under 40 °C for 20 min. To measure Taxol production, 100 µl

reaction volume contained 500 µM DT, 500 µM acetyl-coenzyme A, and 0.5 mg ml$^{-1}$ DBAT in 50 mM sodium acetate buffer, pH 5.5. The reaction was performed under 37.5 °C for 180 min. Reactions were terminated as described above. One unit of activity was defined as the amount of enzyme that catalysed the formation of 1 nM baccatin III or Taxol per minute. The kinetic data on DT were processed via a proportional weighted fit using a nonlinear regression analysis program based on Michaelis–Menten enzyme kinetics[28,54]. Briefly, the kinetic parameters were determined against DT in a concentration range of 15.625–2,000 µM, maintaining acetyl-CoA at a saturation concentration of 2,000 µM. Reactions were conducted under the aforementioned conditions. To set up the standard curves and linear regression equations, different concentrations of baccatin III, DT, and Taxol were prepared respectively: 2, 1, 0.2, 0.1, 0.04, and 0.008 (mg ml$^{-1}$). The HPLC conditions were as described above. The linear relationship between the concentration (mg ml$^{-1}$) and peak area of baccatin III, DT and Taxol was respectively analysed and each equation of the best-fit lines ($R^2 > 0.99$) was determined. The product yield was calculated according to the standard curves (Supplementary Table 11). Values were expressed as the means ± s.d. ($n = 3$). *P* values were calculated by using a two-tailed Student's *t*-test. $P < 0.05$ was considered statistically significant.

**Homology modelling of DBAT three-dimensional structure.** The protein sequences of DBATs from *Taxus* species were retrieved from the NCBI Protein Sequence Database (http://www.ncbi.nlm.nih.gov/protein). Protein Structure Database (Protein Data Bank, PDB) (http://www.rcsb.org/pdb/) provided the template structure of hydroxycinnamoyl-CoA shikimate/quinate hydroxycinnamoyltransferase [*Coffea canephora*] (GenBank accession: ABO47805.1, PDB ID: 4G22), and BAHD family proteins with known structures: hydroxycinnamoyltransferase [*Sorghum bicolor*] (PDB ID: 4KE4), hydroxycinnamoyl-CoA:shikimate hydroxycinnamoyl transferase [*Panicum virgatum*] (PDB ID: 5FAL), vinorine synthase [*Rauvolfia serpentina*] (GenBank accession: CAD89104.2, PDB ID: 2BGH), and anthocyanin malonyltransferase homologue (*Chrysanthemum x morifolium*) (GenBank accession: BAF50706.1, PDB ID: 2E1U). The three-dimensional (3D) model of DBAT protein was built by using SWISSMODEL (http://swissmodel.expasy.org/)[55]. To evaluate the accuracy of the model, PROCHECK was employed[42].

**L-alanine scanning and saturation mutagenesis.** The primers with the single site mutation are listed in Supplementary Table 12. The recombinant plasmid pCWori-*dbat* was employed as a template. The L-alanine scanning mutations and saturation mutations were all conducted by using whole-plasmid amplification PCR[56]. PCR was performed with Phusion DNA polymerase, and the temperature program consisted of 98 °C for 30 s; 25 cycles of 10 s at 98 °C, 30 s at 55 °C, 3 min 20 s at 72 °C, and a final 10 min extension at 72 °C. The PCR products were purified and digested by *Dpn* I at 37 °C for 4 h and then transformed into JM109 competent cells. After the sequences were confirmed, the corresponding clones were used for the enzymatic activity analysis as mentioned above.

**One-pot reaction system.** To select the proper concentration of the DBAT mutant in the reaction system, 0.5, 1.0, 1.5 and 2.0 mg ml$^{-1}$ enzyme solutions, each with 2 mM DT in 5% DMSO (v/v) as the substrate and 2 mM acetyl-CoA as the acetyl donor, in 500 µl reaction volume were respectively tested under the optimum conditions of 37.5 °C and pH 5.5. To compensate for the enzyme activity lost during the reaction, the DBAT mutant supplementing strategy was used at interval of 3 h. To select the proper concentration of the β-xylosidase LXYL-P1-2 in the reaction system, 0.25, 0.5, 1.0 and 1.5 mg ml$^{-1}$ enzyme solutions, each with 2 mM XDT in 5% DMSO (v/v) as the substrate, in 500 µl reaction volume were respectively tested under the optimum conditions of the DBAT mutant. The XDT conversion rate was calculated according to the literature[28]. The reactions were conducted for as long as 15 h. An aliquot of 50 µl sample solution was generally taken at 3, 6, 9 and 12 h respectively for HPLC analysis.

Finally, the ideal concentrations of the DBAT mutant and LXYL-P1-2 were combined to construct the one-pot reaction system with 2 mM XDT as the substrate and 2 mM acetyl-CoA as the acetyl donor. The reaction volume was scaled up from 1 ml through 10 to 50 ml.

**Data availability.** The homology model for DBAT was deposited to the Swiss-Model with accession code Q9M6E2. All other data that support the findings of this study are available within the article and its supplementary files and from the corresponding author upon reasonable request.

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

## Acknowledgements

We thank D. Lu and Y.S. Guo in our Institute for their help in the protein homology modelling, and Y. Li in our Institute for the in vitro cytotoxic detection of Taxol, DT and XDT (with several tumour cell lines). We also thank Y.J. Yuan's lab (Tianjin University, China) for synthesizing the dbat genes. This work was supported by the National Natural Science Foundation of China (Grant nos 81573325, 31270796 and 30770229), the National Mega-project for Innovative Drugs (Grant no. 2012ZX09301002-001-005), and the Youth Fund of Peking Union Medical College (Grant no. 3332015137).

## Author contributions

B.-J.L. designed and performed the experiments and drafted the manuscript. H.W. helped to prepare the mutant proteins. T.G. helped to analyse the enzymatic reaction results. T.-J.C. helped to prepare LXYL-P1-2 protein and make molecular docking. J.-J.C. and J.-L.Y. helped to analyse the data. P.Z. conceived, designed and supervised the study and revised the manuscript. All authors approved the final manuscript.

**Additional information**

**Competing interests:** The authors declare no competing financial interests.

**DOI: 10.1038/ncomms16221**

# Author Correction: Improving 10-deacetylbaccatin III-10-β-O-acetyltransferase catalytic fitness for Taxol production

Bing-Juan Li, Hao Wang, Ting Gong, Jing-Jing Chen, Tian-Jiao Chen, Jin-Ling Yang & Ping Zhu

*Nature Communications* 8:15544 doi: 10.1038/ncomms15544 (2017); Published 18 May 2017, Updated 13 Jul 2018

In the originally published version of this Article, financial support was not fully acknowledged. The PDF and HTML versions of the Article have now been corrected to include support from the National Natural Science Foundation of China grant number 81573325.

