## [Peer Review File · Nature Communications]

Reviewer #1 (Remarks to the Author)

This manuscript makes a commendable effort to readdress a decades old effort to increase taxol yields by converting abundant taxol side products (7-xylulose taxol and 10-deacylated versions) to the target compound. The authors cite these earlier works appropriately.

That said, this reviewer recommends that the authors rework the current article to de-emphasize the notion that their work describes novel activity of DBAT and xylosidases with taxanes. DBAT has been characterized and its substrate scope, including 10-deacetyl taxols with regards to the current work, has been done. Several early attempts to deglycosylate taxols were cited by the authors and thus reduces the novelty of the current work, IF emphasis is placed on enzyme discovery.

Emphasis should be placed mostly on directed evolution methods (there are several examples in the literature with other enzymes) to increase enzyme fitness by the popular method of alanine scanning mutagenesis. Consequently, this will guide the authors to cite references that show successes in these approaches.

There are several instances where the phrasing of sentences needs work; otherwise, the meaning is lost or the flow of the reading is disrupted.

Attachments are included showing annotations within the text that need to be address before this can be a competitive manuscript.

Reviewer #2 (Remarks to the Author)

Li et al. showed an alternative reaction for Taxol production. The alternative reaction does not consume natural sources as the reactants do not solely depend on the Taxus species and is greener in the sense that it produces less waste compared with the conventional production methods. Also, through use of molecular modelling and mutagenesis experiments, authors have showed a double mutant of DBAT enzyme with enhanced catalytic efficiency. Lastly, they used the modeling studies to propose a molecular mechanism for the DBAT enzyme and also its mutant. To my view, this study holds an advancement to the current literature highlighting the need for efficient production of drugs from the natural sources. Hence, it is suitable for publication. However, I would like list one major and a few minor points that should be answered/ revised before publication.

The major shortcoming of the paper is the reliability of the homology model. Authors have used their structural model to interpret the function/selectivity. To be sure about the structure and function relationship, the structural information has to be originated from a physical source like X-ray, or at least the modeling should be defended well enough to attribute functional roles to it. As the authors have stated that the sequence homology between the template and target is only 30%. Percent homology lower than 30% is very close to the twilight zone of alignments (Fischer, Barret et al. 1999). Swiss-model server has straightforward (automated) online modeling applications that works best if the alignment is not in the twilight zone. In this case, although having pivotal importance for the modeling, the pairwise sequence alignment between the template and target is missing. It might be necessary to see how the overall sequences diverge/converge in the alignment. Hence, I would recommend authors to show pairwise sequence alignment used for the modeling step. Also justify why this modeling can be trusted to deduce conclusions on the functionality.

Minor points

Figure 8 is not clear. The ligand, in particular, cannot be distinguished. For all of the parts of Fig. 8 I recommend a revision to clarify the residues and the ligand. Moreover, as suggested by the authors the enhanced catalytic activity obtained for the mutant G38R is likely to be due to two factors. First one is a single hydrogen bond addition to the overall structure, formed by N35 and R38. The second one is the steric clash generated in case of G38 compared with R38. Although single hydrogen bond is very weak (also we cannot assess the distance from the figure) to alter

catalytic efficiency greatly, it may still be reasonable to think that the formation of this H-bond made G38R is more active than the wild-type. However, arginine is much bulkier than glycine. Hence, it is highly unlikely to accept that G38 presents a steric clash, but R38 does not. In other words, the mutagenesis of G38R does not provide any evidence regarding a broader substrate entrance for the mutant. On the contrary, F301V as authors have claimed presents a bulkier space than the wild type. Indeed, F and V have comparable volumes. Overall I would recommend authors to quantitatively investigate the changes in the geometry of the catalytic cleft and then make inferences. Many computational methods, and also web servers, are available and might present a fast and simple alternative for volume calculations.

References

Fischer, D., C. Barret, K. Bryson, A. Elofsson, A. Godzik, D. Jones, K. J. Karplus, L. A. Kelley, R. M. MacCallum, K. Pawowski, B. Rost, L. Rychlewski and M. Sternberg (1999). "CAFASP-1: critical assessment of fully automated structure prediction methods." *Proteins Suppl* 3: 209-217.

Reviewer #3 (Remarks to the Author)

In the manuscript entitled 'Semi-rational site-directed mutagenesis of 10-deacetylbaaccatin III-10- β -O-acetyltransferase (DBAT) and biosynthesis of taxol from 7- β -xylosyl-10-deacetyltaaxol by "one-pot reaction"', the authors describe the possibility of transforming 7- β -xylosyl-10-deacetyltaaxol into taxol by means of the action of the enzyme 10-deacetylbaaccatin III-10- β -O-acetyltransferase. This enzyme controls the formation of baaccatin III from 10-deacetylbaaccatin III, apart from the preliminary removal of the xylosyl group. This interesting study explores the possible use of heterologous systems for obtaining taxol from the related compound 7- β -xylosyl-10-deacetyltaaxol. The manuscript is clearly written and the results are well discussed. The authors have assayed different DBATs obtained from several *Taxus* species and have used the L-alanine scanning mutagenesis methodology to increase the activity of the enzymes studied. For all these reasons, I think this manuscript deserves publication in *Nature Communications*, after the following minor points are amended.

- 1) Why do the authors say that DT is a "by-product"? Is the biosynthesis of this compound known? Do the authors think that DT is formed from taxol? This point should be clarified and expanded on in the manuscript.
- 2) On page 14/23, line 289: Why do the authors assume that the gene expression levels were not different among different species? In which conditions? This assumption is erroneous.
- 3) The strategy used for changing amino acids in order to increase enzyme activity should be supported with more up-to-date bibliography.
- 4) There are a number of minor typing and language mistakes that should be corrected. For example, in the Introduction, page 3/23, line 52: "it" should be "which" ("...which can also be obtained"); in Results, page 5/23, line 127: delete "in" before "consistent"; page 17/23, line 363: *canadiensis* should not begin with a capital letter. A careful reading of the text is necessary to eliminate other errors.

Reviewer #1 (Remarks to the Author):

Q: That said, this reviewer recommends that the authors rework the current article to de-emphasize the notion that their work describes novel activity of DBAT and xylosidases with taxanes. DBAT has been characterized and its substrate scope, including 10-deacetyltaxols with regards to the current work, has been done. Several early attempts to deglycosylate taxols were cited by the authors and thus reduces the novelty of the current work, IF emphasis is placed on enzyme discovery.

A: Thank you for your comments. We have reorganized this part to de-emphasize the notion on the novel activity of DBAT and the β -xylosidase. In the revised manuscript, we just introduced the general research background of the two kind enzymes to readers for their easier understanding of this work.

Q: Emphasis should be placed mostly on directed evolution methods (there are several examples in the literature with other enzymes) to increase enzyme fitness by the popular method of alanine scanning mutagenesis. Consequently, this will guide the authors to cite references that show successes in these approaches.

A: In the revised edition, the emphasis has been placed mostly on directed evolution methods to increase enzyme fitness by alanine scanning mutagenesis, and more references (e.g. Ref. 50-52) including the successful examples have been cited.

Q: There are several instances where the phrasing of sentences needs work; otherwise, the meaning is lost or the flow of the reading is disrupted.

A: We are very sorry for the phrasing and grammar errors in the manuscript, and these errors have been corrected.

Q: Attachments are included showing annotations within the text that need to be addressed before this can be a competitive manuscript.

A: Thank you very much for your help in revision of the text and annotations, we have followed the annotations in the text to re-write or address them (see below and the revised manuscript).

Main questions or annotations in the text:

Q1: In what source (Taxol is about 0.02%, XDT is up to 0.5%)?

A: Line 16 (in the revised manuscript), “in yew trees” was added; line 17, “in the plants” was added.

Q2: By what method? To what method are the authors referring (XDT can be converted into Taxol by de-glycosylation and acetylation)?

A: Line 18-19, “by chemical or biological methods” was added.

Q3: This is NOT a new discovery (see ref 29).

A: The elucidation work was de-emphasized in the revised manuscript.

Q4: What is a “sent coat”?

A: Line 39, according to the description in the ref., the sentence was changed into “for which Taxol is used for the coronary stent or balloon catheter coating”, and more ref. were added.

Q5: In what other genus species is it produced?

A: Besides some endophytic fungi [see ref.: Stierle A. *et al.* Taxol and taxane production by *Taxomyces andreanae*, an endophytic fungus of Pacific Yew. *Science* (1993) 260:214–216. But it is a controversial issue (see ref.: Heinig, U. *et al.* Getting to the bottom of Taxol biosynthesis by fungi. *Fungal Diversity* (2013) 60:161-170)], hazelnut was also reported to produce Taxol [ref.: Ottaggio, L. *et al.* Taxanes from shells and leaves of *Corylus avellana*. *J. Nat. Prod.* (2008) 71:58-60].

Q6: To what are the authors referring, T and pH or just pH?

A: Line 130, the sentence has been revised to define the optimum pH.

Q7: There are no points between 0 and 3h. How could a proper inflection point outside the true linear range be determined. More points are needed between 0-3h.

A: Thank you for your advice, actually, more points had been detected around 3h, and these points have been added in Figure 6a.

Q8: Why did the authors assume the enzyme activity was degrading as the rate approached what could be equilibrium? These acyltransferase reactions are reversible!

A: In the pre-test, we found that the enzyme was not much stable, especially at a relatively higher temperature, and the linear portion of the time course plot was within 3 h, so, we chose the time point of 3 h to initiate the enzyme supplementing to maximize the Taxol yield.

Q9: Why was the concentration chosen? Why not 1.5 mg/mL?

A: Line 278-279, the sentence has been reorganized: “All of the yields met the requirement for the next reaction from DT to Taxol and the relatively lower concentration of 0.5 mg/mL LXYL-P1-2 was chosen in the following “one-pot reaction” system”. It is also to limit the enzyme consumption.

Q10: The entire 50 mL lot was purified on the HPLC?

A: Yes, at first, the product from the entire 50 mL was extracted through ethyl acetate, then, the product was concentrated into a small volume, and finally, the product was purified and prepared by HPLC.

Q11: How many (steps are there in Taxol biosynthesis)?

A: Line 297-302, the sentences have been reorganized. From cited ref., there are 20 enzymes involved in 19 steps of the biosynthetic pathway from GGPP to Taxol.

Other changes have also been made mainly based on the instructions of the reviewer (see the revised manuscript).

Reviewer #2 (Remarks to the Author):

Q: The major shortcoming of the paper is the reliability of the homology model. Authors have used their structural model to interpret the function/selectivity. To be sure about the structure and function relationship, the structural information has to be originated from a physical source like X-ray, or at least the modeling should be defended well enough to attribute functional roles to it. As the authors have stated that the sequence homology between the template and target is only 30%. Percent homology lower than 30% is very close to the twilight zone of alignments (Fischer, Barret et al. 1999). Swiss-model server has straightforward (automated) online modeling applications that works best if the alignment is not in the twilight zone. In this case, although having pivotal importance for the modeling, the pairwise sequence alignment between the template and target is missing. It might be necessary to see how the overall sequences diverge/converge in the alignment. Hence, I would recommend authors to show pairwise sequence alignment used for the modeling step. Also justify why this modeling can be trusted to deduce conclusions on the functionality.

A: Thank you for your comments and suggestions. Since the crystal structure of DBAT is not available, the homology modeling has to be used in this work. Generally, the term “homology” may not be measured by percentage, you probably mean the “similarity”. It’s our fault that we did not involve these very important data in the original manuscript. These data have been added in the revised manuscript. The sequence similarity between DBAT and the template was 45%, which should guarantee the accuracy of the predicted structure. Thank you for your suggestion, the pairwise sequence alignment between the template and DBAT has been added to Fig. 3 and in the supplementary materials, which made the article more readable. And the corresponding paragraphs have been reorganized with more ref. The predicted model was assessed to be qualified by a Ramachandran plot (Supplementary Fig. 3). DBAT also showed similar hydrophobicity and distribution of electronegativity residues with the template (Supplementary

Fig. 4). Therefore, we think that the homology model is reasonable for the protein engineering research of DBAT, although it may have some shortcomings.

Minor points

Q: Figure 8 is not clear. The ligand, in particular, cannot be distinguished. For all of the parts of Fig. 8 I recommend a revision to clarify the residues and the ligand.

A: Fig. 8 has been revised to highlight the ligand and the two mutation sites.

Moreover, as suggested by the authors the enhanced catalytic activity obtained for the mutant G38R is likely to be due to two factors. First one is a single hydrogen bond addition to the overall structure, formed by N35 and R38. The second one is the steric clash generated in case of G38 compared with R38. Although single hydrogen bond is very weak (also we cannot assess the distance from the figure) to alter catalytic efficiency greatly, it may still be reasonable to think that the formation of this H-bond made G38R is more active than the wild-type. However, arginine is much bulkier than glycine. Hence, it is highly unlikely to accept that G38 presents a steric clash, but R38 does not. In other words, the mutagenesis of G38R does not provide any evidence regarding a broader substrate entrance for the mutant.

A: Thank you for your reminding. In the revised manuscript, we did not emphasize the single hydrogen bond formation between N35 and R38, since further analysis showed that the H bond was also formed between N35 and G38 in the wild type; and you are right, arginine is much bulkier than glycine, there must be other reasons for the improved catalytic efficiency of the G38R mutation. We found that the amino acid residue of 38 site is near the surface, and the side-chain of Arg is more hydrophilic than that of Gly, so the increased hydrophilic property of G38R mutation is probably one of the reasons for the elevated catalytic efficiency. The related paragraphs have been reorganized in the revised manuscript.

Q: On the contrary, F301V as authors have claimed presents a bulkier space than the wild type. Indeed, F and V have comparable volumes. Overall I would recommend authors to quantitatively investigate the changes in the geometry of the catalytic cleft and then make inferences. Many computational methods, and also web servers, are available and might present a fast and simple alternative for volume calculations.

A: According to the reviewer's suggestion, we have further analyzed the structure of F301V mutant, including the volumes of phenylalanine and valine. And the paragraph (Line 367-389) has been reorganized.

Reviewer #3 (Remarks to the Author):

Q: Why do the authors say that (X)DT is a "by-product"? Is the biosynthesis of this compound known? Do the authors think that (X)DT is formed from taxol? This point should be clarified and expanded on in the manuscript.

A: Thank you for your comments. During the extraction process, XDT is often obtained as a companion of Taxol, so it is defined as a by-product of Taxol. How XDT is formed is unknown, and we have discussed this question in the revised manuscript (Line 306-310).

Q: On page 14/23, line 289: Why do the authors assume that the gene expression levels were not different among different species? In which conditions? This assumption is erroneous.

A: Thank you for your reminding. We have reorganized this sentence (Line 321-323).

Q: The strategy used for changing amino acids in order to increase enzyme activity should be supported with more up-to-date bibliography.

A: We have cited more up-to-date bibliography about protein engineering (Ref 50-52).

Q: There are a number of minor typing and language mistakes that should be corrected. For example, in the Introduction, page 3/23, line 52: “it” should be “which” (“...which can also be obtained”); in Results, page 5/23, line 127: delete “in” before “consistent”; page 17/23, line 363: *canadiensis* should not begin with a capital letter. A careful reading of the text is necessary to eliminate other errors.

A: Thank you for your careful examination. We have rechecked and changed all the typing and language errors.

Again, thank you for your kind letter and the constructive comments.

Reviewer #1 (Remarks to the Author)

Lines 22-23: Reword: "The research not only exhibited potential substrate binding or catalytic sites of DBAT..."

This is a vague remark.

Rewrite: "In this study, the structure of an enzyme homolog was used to model DBAT, identify its active site, and to design a double mutant DBAT(G38R/F301V) that had a k_{cat}/K_m of XXXX."

L25: Change to: "An in vitro..."

L25: Omit quotation marks around "one-pot reaction"...unnecessary. (CHANGE ALL instances of this)

L28: Change to: ...another choice for biocatalytic production of Taxol...

L48: Change to: ..."percentage found in the bark.9"

L64: Xylosyl (capital 'X' at start of sentence)

L64: Explain here very briefly why XDT is considered a by-product and not a precursor of Taxol.

L65-66: Change to: "...is even higher than that of Taxol in nursery cultivated..." AND "This manufacturing by-product is usually..."

L84: "These efforts..."

L92: "...cloned the DBAT..."

L97-112: this was rewritten nicely

L203: Add reference at end of sentence "...mutated to Ala).(REF)"

L236 and 239: Superscript "/"

Figure 6a: adding time points for 1.5 mg/mL curve <3 h is sufficient.

L327: "...under native or natural (CHOOSE ONE) conditions..."

L330: "this study modeled a 3D structure of DBAT on the structure of...HCT...as the template..."

L402: "the catalytic efficiency..."

L404: "since the catalytic..."

L408: "3-h..." (hyphen needed)

L409: "1-mL..." (hyphen needed)

Reviewer #2 (Remarks to the Author)

Authors' response in the defense of the homology model is valid. If homology is present, sequence similarity higher than 30% is acceptable for modeling.

I would like point out a few points regarding the newly added alignment figure. Such that V and F

are accepted as similar to M, however F is not found similar with V. Is there any particular choice of authors for definition of amino acid similarity? Additionally, some other parts of the alignment might be problematic. Particularly, P37 in DBAT, instead of L36, was aligned with a leucine of the template. Similarly, (around the position 180) E in DBAT, instead of its preceding G, was aligned with a G in the template. These parts although seemed subtle but might be resulted from a wrong scoring scheme etc and can affect the whole nature of the alignment itself. Another point is that the residues that are marked with red stars and mentioned to be involved in substrate interaction are not conserved at all, more specifically 2 of them are absent or deleted in the template. Do authors provide any particular biological explanation for such variation? Or is it an outcome of the selected parameters of alignment such as gap penalties and substitution matrices. Quality of the alignment should be further checked and justified.

Lastly, authors' revision regarding the geometry of the catalytic cleft are acceptable.

Reviewer #3 (Remarks to the Author)

The authors have modified the manuscript according to all my suggestions and have improved the language. I therefore consider that the new version of the manuscript is suitable for publication. Only one aspect needs to be corrected: the abstract should be separated from the introduction.

Reviewer #1 (Remarks to the Author):

Q: Lines 22-23: Reword: “The research not only exhibited potential substrate binding or catalytic sites of DBAT...”

This is a vague remark.

Rewrite: “In this study, the structure of an enzyme homolog was used to model DBAT, identify its active site, and to design a double mutant DBAT(G38R/F301V) that had a k_{cat}/K_m of XXXX.”

L25: Change to: “An in vitro...”

L25: Omit quotation marks around “one-pot reaction”...unnecessary. (CHANGE ALL instances of this)

L28: Change to: ...another choice for biocatalytic production of Taxol...

L48: Change to: ...”percentage found in the bark.”

L64: Xylosyl (capital ‘X’ at start of sentence)

L64: Explain here very briefly why XDT is considered a by-product and not a precursor of Taxol.

L65-66: Change to: “...is even higher than that of Taxol in nursery cultivated...” AND “This manufacturing by-product is usually...”

L84: “These efforts...”

L92: “...cloned the DBAT...”

L97-112: this was rewritten nicely

L203: Add reference at end of sentence “...mutated to Ala).(REF)”

L236 and 239: Superscript “/”

Figure 6a: adding time points for 1.5 mg/mL curve <3 h is sufficient.

L327: “...under native or natural (CHOOSE ONE) conditions...”

L330: “this study modeled a 3D structure of DBAT on the structure of...HCT...as the template...”

L402: “the catalytic efficiency...”

L404: “since the catalytic...”

L408: “3-h...” (hyphen needed)

L409: “1-mL...” (hyphen needed)

A: We have revised all of them according to the reviewer’s instruction. Since the term “by-product” was not the key point in this manuscript, we deleted it in the revised version, only using the term “analogue” for XDT, and we have briefly explained here why XDT is considered not a precursor of Taxol. Thank you very much for polishing our manuscript.

Reviewer #2 (Remarks to the Author):

Q: Authors' response in the defense of the homology model is valid. If homology is present, sequence similarity higher than 30% is acceptable for modeling.

I would like point out a few points regarding the newly added alignment figure. Such that V and F are accepted as similar to M, however F is not found similar with V. Is there any particular choice of authors for definition of amino acid similarity? Additionally, some other parts of the alignment might be problematic. Particularly, P37 in DBAT, instead of L36, was aligned with a leucine of the template. Similarly, (around the position 180) E in DBAT, instead of its preceding G, was aligned with a G in the template. These parts although seemed subtle but might be resulted from a wrong scoring scheme etc and can affect the whole nature of the alignment itself. Another point is that the residues that are marked with red stars and mentioned to be involved in substrate interaction are not conserved at all, more specifically 2 of them are absent or deleted in the template. Do authors provide any particular biological explanation for such variation? Or is it an outcome of the selected parameters of alignment such as gap penalties and substitution matrices. Quality of the alignment should be further checked and justified.

Lastly, authors' revision regarding the geometry of the catalytic cleft are acceptable.

A: Thank you for your comment.

A: Thank you for your comments. You are right, in the Figure 3a of second version of the manuscript, P37 in DBAT, instead of L36, was aligned with a leucine of the template, and similar phenomenon was found in the alignment of E184 of DBAT with a glycine of the template. But, this blast result was given by using the software Clustal X. In the final version of the manuscript, the G183 has been corrected to align with a Gly in the template based on the NCBI-Blast and DNAMAN alignment results. Unfortunately, again, the L36 in DBAT was not aligned with a leucine in the template by the other softwares. By the way, the revised result does not affect the identity and similarity of the two sequences. And, in fact, the alignment result does not mean F is similar with V.

Secondly, the information that the residues marked with red stars were involved in substrate point was from the alignment of other BAHD family members, which was given by other authors. Since compared with BAHD other members, DBAT has a low sequence identity and varied substrates, the substrate binding sites may be varied.

Thank you again for your time.